# Investigating Runway Incursion Incidents at United States Airports

Olajumoke Omosebi [1], Mehdi Azimi [1], David Olowokere [2], Yachi Wanyan [2], Qun Zhao [1] and Yi Qi [1,*]

1    Department of Transportation Studies, Texas Southern University, Houston, TX 77004, USA;
o.omosebi1369@student.tsu.edu (O.O.); mehdi.azimi@tsu.edu (M.A.); qun.zhao@tsu.edu (Q.Z.)
2    Department of Engineering, Texas Southern University, Houston, TX 77004, USA;
david.olowokere@tsu.edu (D.O.); yachi.wanyan@tsu.edu (Y.W.)
*    Correspondence: yi.qi@tsu.edu; Tel.: +1-713-313-6809

**Abstract:** According to the Federal Aviation Administration (FAA), the number of runway incursions is increasing. Over the last two decades, the number of runway incursions at U.S. airports has increased from 987 in 2002 to 25,036 in 2020. Runway incursions are a major threat to aviation safety, causing major delays and financial consequences for airlines, as well as injury or death through incidents such as aircraft collisions. The FAA promotes the implementation of runway safety technology, infrastructure, procedural methods, alterations to airport layouts, and training practices to reduce the frequency of runway incursions. In this paper, the relationship between airport geometry factors, mitigating technologies, and the number of runway incursions at large hub airports in the United States was investigated using a random effects Poisson model for analyses of panel data. Airport operations data from the FAA Air Traffic Activity System, runway incursion data from the FAA Aviation Safety Information Analysis and Sharing System from 2002 to 2020, and airport geometry data created using airport geometry features from the FAA airport diagrams were collected. Thirty large hub airports with FAA-installed mitigating technologies were investigated. The model identified significant variables that correlate with runway incursions for large hub airport categories defined by the National Plan of Integrated Airport Systems (NPIAS). The model results indicate that airports with significant numbers of runway-to-runway intersection points increase runway incursion rates and mitigating technologies Runway Status Lights (RWSLs) and Airport Surface Detection Equipment, Model X (ASDE-X), can help reduce runway incursions at severity levels A and B.

**Keywords:** runway incursions; airport geometry; ASDE-X; RWSL; airport safety; panel model

## 1. Introduction

In 2007, the FAA adopted the definition of the International Civil Aviation Organization (ICAO) that a runway incursion is "any incident at an airport involving the improper presence of a vehicle, aircraft, or person on a protected surface intended for aircraft landing and taking off." [1] Aviation. Severity levels are divided into four categories, with A being the most severe and D being the least severe [1], as follows:

- Category A: An occurrence that was exceedingly serious and narrowly escaped a collision.
- Category B: A circumstance in which there is a high chance of a collision; to avoid one, a quick corrective action or evasive maneuver may be required.
- Category C: An occurrence in which there was enough time or space to escape a collision.
- Category D: An incident that fits the criteria for a runway incursion but has no immediate safety repercussions as a result of an incorrectly parked car, person, or aircraft on a runway or in a protected location.

In addition, the following categories of factors that contribute to runway incursions are used by the FAA and ICAO to categorize incursions [2]:

Operational incident (OI): a decision made by an air traffic controller that causes less space than what is necessary between two or more aircraft or between an aircraft and an obstruction.

Pilot deviation (PD): a pilot's activity that contravenes a Federal Aviation Regulation, such as entering the runway without authorization.

Vehicle/pedestrian deviation (V/PD): the unauthorized entry of vehicles or pedestrians into airport movement areas without the consent of the air traffic controller (ATC).

Reducing runway incursions is one of the FAA's top priorities. For more than a decade, avoiding runway incursions has been on the FAA's "Most Wanted" list of safety improvements. Although the FAA has been interested in reducing runway incursions for many years, recent incidents demonstrate that runway incursions are still an issue. The number of runway incursions has been increasing since 2002, with 26,357 runway incursions occurring between 2002 and 2020 [2]. Evidence has indicated that as traffic volume increases, so does the number of runway incursion accidents. The runway incursion rate climbed by more than 43% between 1988 and 1990, while the volume of travel at towered airports in the U.S. also rose by 4.76% [3]. The RI rate decreased by 30% between 1990 and 1993, a period during which the volume of traffic decreased by 5.34% [3]. Although the number of runway incursions that end in accidents is quite low, in the past ten years there has been no discernible decline in the frequency of these incidents. Figure 1 depicts the surge in runway incursions from 2010 to 2018, with an increase of almost 85% in overall incursions.

**Figure 1.** Runway incursions from 2010 to 2018.

Runway incursions are a very complicated issue and can be investigated from different perspectives. Some previous studies investigated the human factors associated with runway incursions [4–6]. Some studies investigated the factors that contribute to OI incursions [7–9]. Some studies have been conducted on identifying the impacts of airport geometry characteristics on runway incursions [10–12]. Recently, the FAA invested in a range of technology-based runway mitigating technologies, including Airport Surface Detection Equipment, Model X (ASDE-X), and Runway Status Lights (RWSLs). Studies have been conducted on investigating the effectiveness of these technologies [13–16]. However, most previous studies only investigated one technology with a small number of airports and a range of a few years. In addition, some findings from these previous studies are inconsistent. For example, Schonefeld and Moller [13] found that the implementation of RWSLs reduced incursions at an airport by up to 70%. However, Ison [16] concluded that

the RWSL technology did not contribute to a reduction in runway incursions at two studied airports. Croft (2015) [14] found that the use of ASDE-X resulted in a reduction in the number of Category A and B incursions. However, Claros et al. [15] indicated that despite the implementation of ASDE-X technology, seven out of the nine airports studied did not experience reduced severe runway incursions. Due to these inconsistencies in the literature, more studies are needed to further investigate the impacts of these technologies on runway incursions based on a large sample dataset.

To fill this research gap, this study investigated runway incursions at 30 commercial airports in the United States' large hubs based on data collected over 19 years (from 2002 to 2020). Also, for this study, we investigated the runway incursion problem from the engineering perspective by focusing on the impacts of airport geometries and mitigating technologies. Thus, the research questions that guided this study were as follows:

- Research question 1: do runway incursion mitigation technologies installed at an airport contribute to reducing the number of runway incursions at the airport?
- Research question 2: do the airport runway geometric characteristics of an airport contribute significantly to the number of runway incursions at the airport?

The results of this research will assist the aviation industry in better understanding the relationship between runway geometries, mitigating technologies, and runway incursions to potentially reduce runway incursions.

## 2. Background

Runway incursions have previously resulted in major accidents with considerable casualties. One of the deadliest aviation catastrophes in history was triggered by a runway incursion. For example, in 1977, two Boeing 747s collided in Tenerife, Canary Islands, Spain, killing 583 people (National Transportation Safety Board). Severe runway incursions from 1977 to 2022 that caused fatalities and damage to aircraft and airport properties are summarized in Table 1.

**Table 1.** An overview of significant runway incursion incidents from 1977 to 2020 (source: [17]).

| Year | Runway Incursion Occurrence |
|---|---|
| 1977 | In Tenerife, two commercial airplanes collided on the runway, killing 583 people. |
| 1983 | In Madrid, Spain, a runway collision involving two commercial aircraft resulted in 100 fatalities. |
| 1990 | When a North-West Airlines Boeing 727 and a DC-9 crashed on a foggy runway in Detroit, Michigan, eight persons were killed and thirty-six were injured. |
| 1990 | In Atlanta, Georgia, a small twin-engine aircraft that had not taxied off the runwaycollided with a Boeing 727 that was landing and caused one fatality. |
| 1991 | A Boeing 737 was landing at Los Angeles International Airport when it crashed with a commuter plane that was waiting on the runway, killing 34 passengers. |
| 1994 | In St. Louis, Missouri, the occupants of a tiny twin-engine plane were killed when the plane taxied into the path of a DC-9 landing on the same runway. |
| 1996 | 14 people were killed when a twin-engine business plane taxied onto a runway at an unattended airport in Quincy, Illinois, as a commuter plane was touching down. |
| 1999 | A commercial airliner on takeoff came within 300 feet of another commercial airliner that had taxied onto the runway in four consecutive occurrences (two at Chicago O'Hare, one at Los Angeles, and one at JFK in New York). |
| 1999 | Two single-engine private planes collided on a runway near Sarasota, Florida, killing four persons. |
| 2000 | In Taiwan, a Singapore Airlines B-747 took off at night during a typhoon on a blocked runway before colliding with construction machinery and causing the deaths of 82 people. |
| 2012 | During a touch-and-go attempt, a Cessna 172N Ram killed a person mowing the grass at Tone Airfield. |
| 2014 | On takeoff from Moscow Vnukovo Airport, a Dassault Falcon 50 crashed with a snowplow that had strayed onto the runway, killing the CEO and Chairman of the oil company Total, Christophe de Margerie. |
| 2020 | The Austin-Bergstrom International Airport was invaded by an adult male invader who made his way to runway 17R before being hit and killed by a Boeing 737-7H4 Southwest Airlines Flight 1392 as it touched down at the airport. There were no injuries or fatalities among the 58 individuals on board, but the 737's left engine nacelle incurred significant damage. |

To address multi-agency concerns regarding runway incursions, the FAA has invested in a range of technology-based safety measures. For example, an airport surface traffic management system known as Detection Equipment, Model X (ASDE-X), provides air traffic controllers seamless coverage, aircraft identification, and the capacity to spot potential runway conflicts by giving them thorough coverage of the activity occurring on runways and taxiways. In addition, a runway safety technology called Airport Surface Surveillance Capability (ASSC) enables controllers and pilots to identify potential aircraft and ground vehicle runway conflicts on the airport's surface as well as on approach and departure pathways within a short distance of the airport. Also, Runway Status Lights (RWSLs) are technological lights implanted in the runway and taxiway pavement that give immediate, clear notifications without any input from a controller. The signal automatically changes to red when it is unsafe to enter, cross, or start takeoff due to other traffic, allowing pilots and vehicle operators to stop when runways are hazardous. This study will concentrate on the use of RWSLs and ASDE-X in large airports because the FAA has heavily installed RWSLs and ASDE-X at large hubs while using fewer RWSLs at medium hubs and fewer ASSC technologies at small hubs. A system's effectiveness, such as that of ASDE-X and RWSL, can be learned by investigating runway incursions. They can also aid in assessing the return on investment of billions in taxpayer money and in assessing the distribution of safety improvement funding to the best technologies, practices, and educational initiatives [13,18].

### 3. Data Collection

The data for this study ranged from 2002 to 2020. The sampling list for this research consisted of a group of airports with similar characteristics in the same major airport category as the National Plan of Integrated Airport Systems (NPIAS). The National Plan of Integrated Airport Systems (NPIAS) categorizes commercial service airports (large hub, medium hub, small hub, and non-hub) based on annual passenger enplanements, and these airports are eligible to receive federal grants through the Airport Improvement Program (AIP). Of the 396 primary airports in the National Plan of Integrated Airport Systems (NPIAS), 30 are large hubs. Table 2 shows how the FAA categorizes airports based on the number of passengers that board an aircraft at an airport.

**Table 2.** FAA categories of airport activities (*source*: [1]).

| Definition | Criteria | Also Referred to As |
|---|---|---|
| Commercial Service: Publicly owned airports with at least 2500 annual enplanements and scheduled air carrier service. Primary airports are commercial service airports with more than 10,000 annual enplanements. | | |
| Large Hub | Receives 1 percent or more of the annual U.S. commercial enplanements | Primary |
| Medium Hub | Receives from 0.25 to 1.0 percent of the annual U.S. commercial enplanements | Primary |
| Small Hub | Receives from 0.05 to 0.25 percent of the annual U.S. commercial enplanements | Primary |

Large hub airports handle high volumes of commercial traffic, and air traffic controllers play an important role in ensuring safe operations. Countermeasures based on advanced technologies are also most appropriate at large hub airports, where the volume of commercial aircraft and the high cost of even minor delays justify the significant financial outlay required. Operating characteristics, financial resources, infrastructure, and technology deployment can differ between and within airport categories.

All information was obtained from the FAA's Operations Network (OPSNET) and Runway Incursions (RWS) databases. OPSNET was used to collect system-wide airport tower operations volume data. Counts of runway incursions, categorized by severity type, were collected from the Runway Incursions (RWS) database (i.e., A, B, C, and D).

For sampling purposes, a list of airports from the NPIAS database that met the criteria of commercial service, primary, and large hub was generated. The final list of 30 airports included large hub airports with similar characteristics as a result of similar administration practices and funding sources. This particular set of samples was tested. The airports in the sample had minimal changes in airfield geometry and low-impact construction projects that did not have a significant impact on runway and taxiway operations such as taxiway or runway closure. During the study period from 2002 to 2020, airport diagrams, historical aerial photographs, and available press releases or project reports were used to verify airport changes. Table 3 lists the 30 large airports selected from the NPIAS list used in this study.

**Table 3.** List of samples of the large hub airports [2].

| Airport | ID | Airport | ID |
|---|---|---|---|
| Hartsfield-Jackson Atlanta International | ATL | Los Angeles International | LAX |
| General Edward Lawrence Logan International | BOS | Laguardia | LGA |
| Baltimore/Washington International Thurgood Marshall | BWI | Orlando International | MCO |
| Charlotte/Douglas International | CLT | Chicago Midway International | MDW |
| Ronald Reagan Washington National | DCA | Miami International | MIA |
| Denver International | DEN | Minneapolis-St Paul International/Wold-Chamberlain | MSP |
| Dallas-Fort Worth International | DFW | Chicago O'Hare International | ORD |
| Detroit Metropolitan Wayne County | DTW | Portland International | PDX |
| Newark Liberty International | EWR | Philadelphia International | PHL |
| Fort Lauderdale/Hollywood International | FLL | Phoenix Sky Harbor International | PHX |
| Daniel K Inouye International | HNL | San Diego International | SAN |
| Washington Dulles International | IAD | Seattle-Tacoma International | SEA |
| George Bush Intercontinental/Houston | IAH | San Francisco International | SFO |
| John F Kennedy International | JFK | Salt Lake City International | SLC |
| McCarran International | LAS | Tampa International | TPA |

### 3.1. Development of Variables

Based on the data collected from these 30 large hub airports, the following dependent and independent variables were developed for analysis by using the model presented in the methodology section. A brief description of these variables is provided below in Table 4, and detailed explanations of the development of these variables are provided in the following sections.

**Table 4.** Dependent and independent variables.

| Variables | Description |
|---|---|
| *Dependent Variable* | |
| Levels of severity A and B | The total number of runway incursions at severity levels A and B per year at an airport. |
| *Independent Variables* | |
| RWY_RWY | The total number of runway-to-runway intersecting points at an airport. |
| ASDE-X Tech | A runway incursion surface surveillance mitigation technology. |

**Table 4.** *Cont.*

| Variables | Description |
|---|---|
| RWSL Tech | A runway Incursion mitigation technology. |
| Num of RWY | The total number of runways in each airport. |
| Total RWY Length | The addition of the entire runway length. |
| Acute Angle | A taxiway exit turn that is less than 90 degrees. |
| Right Angle | Runway and taxiway turns that are 90 degrees. |
| Crossing TWY | A taxiway that crosses or intersects a runway or taxiway; they are opposite to each other. |
| AirportOperation_100000 | Total annual takeoffs and landings of aircraft at an airport (in 100,000) |
| Single RWY | The single runway at an airport which is used for both takeoffs and landings. |
| Parallel RWY | Two or more runways at an airport whose centerlines are parallel. |
| Mixed Runway | Airports with both single and parallel runways. |
| Crossing Runway | Airports with runways that intersect with other runways. |

### 3.1.1. Dependent Variables

The dependent variable was the total number of runway incursion incidents at severity levels A and B per year at an airport. As we mentioned before, A and B are the most severe runway incursion levels. The dependent variables were collected from the FAA Runway Safety Office Runway Incursions (RWS) database. The incident data appear to contain 4362 records for runway incursion events at 30 large hub airports from 1 January 2002 to 31 December 2020. Data were collected between the years 2002 and 2020. All datasets (2002–2020) were collected in a calendar year from January to December of each year.

### 3.1.2. Independent Variables

The independent variables were selected from three different sections: (1) the Operations Network (OPSNET), (2) airport geometry, and (3) airport mitigating technologies.

1.  *Operations Network (OPSNET)*

First, operations data for the calendar years 2002–2020 for the 30 large hub airports were downloaded from the Operations Network (OPSNET) in a CSV file. Due to the fact that the number of annual airport operations is large, the data were rescaled to per 100,000 operations. The air traffic volume data set provided quantitative estimates of the number of landings and takeoffs (combined) that occurred at each airport each year.

2.  *Airport Geometry*

Second, the independent airport geometry data were derived from a visual examination of the features on each airport layout diagram, which were grouped into two sections: (a) runway variables and (b) taxiway variables. Figure 2 below shows the example airport layout of General Edward Lawrence Logan International (BOS) and provides an example of the airport geometry variables that were collected and utilized in this research.

Note that the table in this figure depicts the description of the runway and taxiway variables that were visually collected from the airport layout diagram. All the layout diagrams of the 30 large hub airports assessed in this study were collected visually in a similar way.

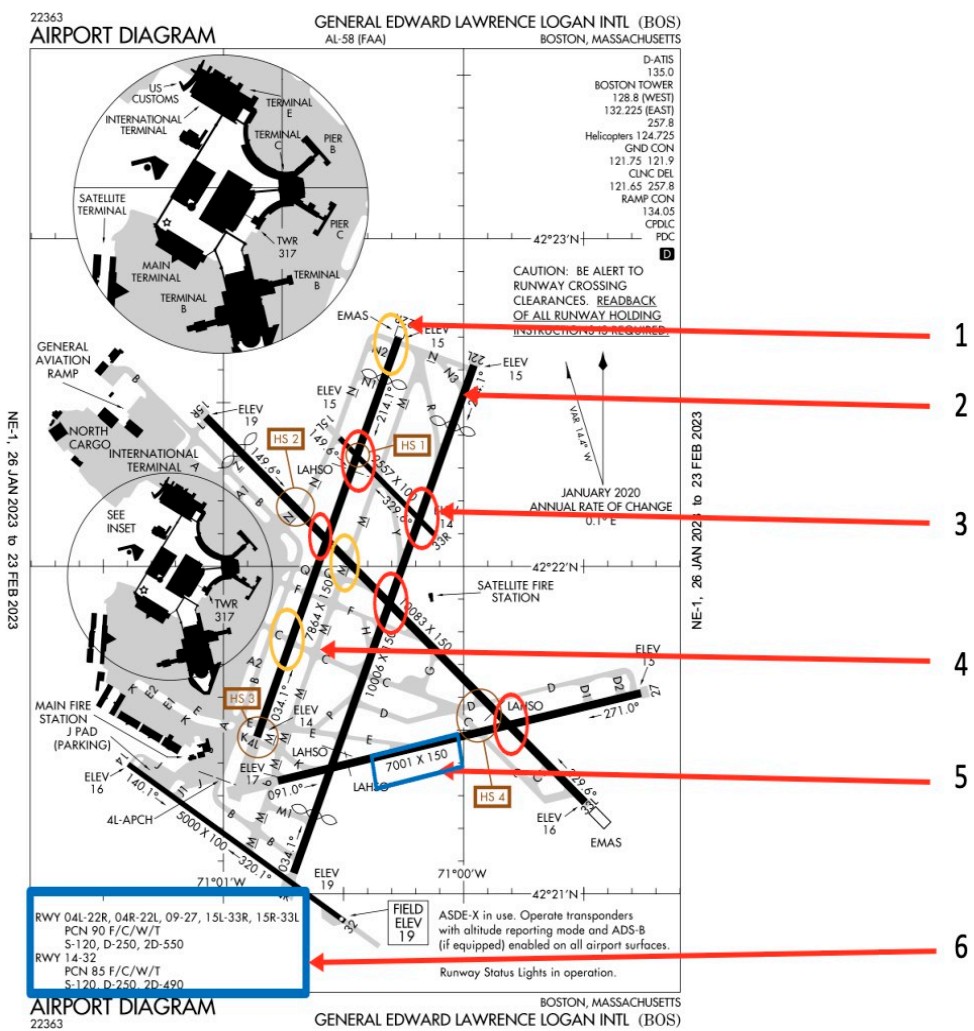

**Figure 2.** Example of an airport layout diagram and the variables that were extracted from the layout.

| Numbers | Symbol | Description |
|---------|--------|-------------|
| 1 | ◯ | Runway to the taxiway intersection point. The grey line intersects with the black line. |
| 2 | ▮ | Black lines represent the runways |
| 3 | ◯ | Runway to a runway intersection point Black line intersects with another black line. |
| 4 | │ | Gray lines represent the taxiways |
| 5 | 7001 x 150 | Runway length and width |
| 6 | RWY 04L-22R,09-27 PCN 90 F/C/W/T S-120, D-250, 2D- | Runways number. Runway number is also always at both end of each runway line. |

*2a. Runway Variables:*

Airports have unique geometric configurations. The runway variables in this study were classified via the runway configuration, which included the type of runway (single, parallel, and mixed runways), the number of runway intersections, the number of runways, and the total runway length. The model's representation of the runway parameters was

based on this classification and the corresponding runway lengths at each airport. For instance, an airport may have both single and parallel runway configurations; therefore, both categories were considered. The number of runways in each category was summed, for example, for parallel runways, the lengths of both runways were added. A mixed runway configuration is the combination of one or more types, such as single and crossing, single and parallel, or single and parallel and crossing.

Figure 3 is an example of the classification of the types of runways that were visually collected from the airport layout diagrams.

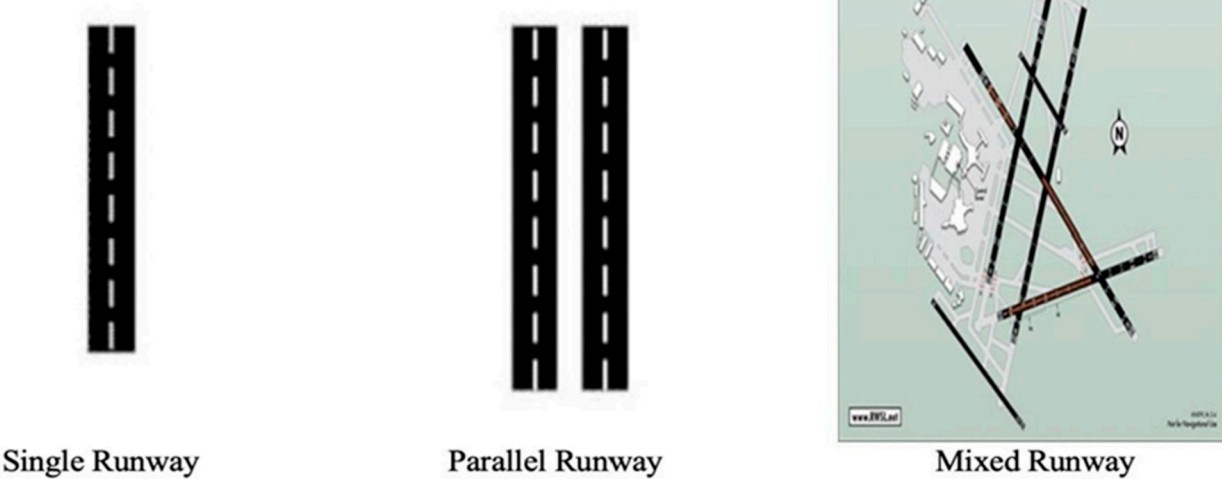

**Figure 3.** Classification of the types of runways.

*2b. Taxiway Variables:*

Taxiways are used to control how aircraft land and depart from the runway. The model considered three different types of taxiway variables: right angle; acute angle, also known as a high-speed taxiway' and crossing taxiway, or two right angles. Figure 4 shows examples of taxiway-to-runway intersections at the airport that may be associated with the rate of runway incursions.

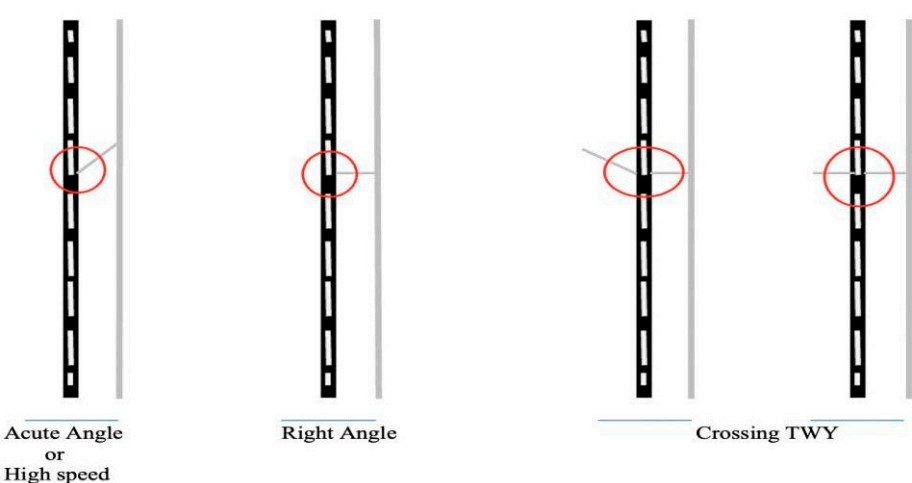

**Figure 4.** Examples of taxiway intersections used in this study.

3.     *Airport-Runway-Incursion-Mitigating Technologies*

Lastly, the initial operating capability dates of the surface technologies (ASDE-X AND RSWL) for each airport were extracted from the FAA report.

For the ASDE-X ground radar, a surveillance system that uses radar and satellite technology to track the surface movements of aircraft and vehicles. It was developed to assist in the reduction of critical category A and B runway incursions.

Runway Status Lights (RWSLs) technology is embedded in runway and taxiway pavement and alerts pilots and vehicle operators when runways are unsafe. When other traffic makes it unsafe to enter, cross, or begin takeoff, the lights automatically turn red. The lights provide direct, immediate alerts and do not require controller input.

## 4. Methodology

In this study, the panel data modeling method was used to analyze the impacts of various factors on the number of runway incursions at an airport.

Panel data refers to samples of the same cross-sectional units observed at multiple points in time. A panel-data observation has two dimensions: $i = 1 \cdots N$ denotes the cross-sectional unit, and $t = 1 \cdots T$ denotes the period of the observation. In this study, the data consist of information collected from 30 large hub airports over 19 years ($N = 30$ and $T = 19$). Therefore, it can be viewed as panel data, and models for panel data can be applied to take into account the heterogeneity in runway incursion frequencies across different airports. Furthermore, since the dependent variable, i.e., the annual runway incursion frequency, is count data, the Poisson regression model for panel data was selected for this study. In general, the Poisson regression model for panel data can be expressed mathematically as follows:

$$f(Y_{it} = y_{it}|X_{it}) = \frac{\exp(-\lambda_{it})\lambda_{it}^{y_{tj}}}{y_{it}!} \tag{1}$$

where $f(.)$ is the probability mass function (pmf) of $Y_{it}$, which is assumed to draw from a Poisson distribution with the parameter $\lambda_{it}$, which is the expectation of $Y_{it}$. By relating $\lambda_{it}$ to independent variables via the following Equation, the independent variables can be incorporated into the model.

$$Exp(Y_{it}) = \lambda_{it} = \exp\left(\beta' X_{it} + \alpha_i + \varepsilon_{it}\right) \tag{2}$$

where $Y_{it}$ is the dependent variable, i.e., the number of runway incursions at an airport $i$ ($i = 1 \cdots 30$) during the year $t$ ($t = 1 \cdots 19$); $X_{it}$ are the vector of independent variables, as listed in Table 4. In the panel data model, there are two types of independent variables: (1) the individual-specific variables which are specific to the airport $i$ and to be constant over time (over the different years), such as the airport geometric characteristics listed in Table 4; and (2) time-variant variables which change over time, such as the airport operation. $\alpha_i$ is the individual effect which is specific to the individual airport $i$ and is constant over time, $\varepsilon_{it}$ is the error term, and $\beta$ is the coefficient vector for $X_{it}$.

In general, there are two types of panel data models: the fixed-effects model and the random-effects model. The random-effects model assumes that the individual-specific effects $\alpha_i$ are distributed independently of the independent variables, while the fixed-effects model allows $\alpha_i$ to be correlated with the independent variables. In the random-effects model, $\alpha_i$ is assumed to be an independently and identically distributed normal random variable with a mean 0 and a variance $\sigma^2$, and in the fixed-effects model, $\alpha_i$ is included as an individual specific intercept for the airport $i$. The standard fixed-effects model cannot identify the effects of any individual–specific variables because it requires within-group variation for the model estimation [19]. Therefore, in this study, the random-effects model was selected.

## 5. Result Analysis

In this study, the random effects Poisson models for panel data were developed using the "pglm" package(Version 0.2-3) in R software(version 4.2.2). Thirteen independent variables were considered in the model, and Table 4 provides definitions and values for

each one. Since many airports that implemented the technology ASDE-X also installed RSWL technology, the independent variables ASDE-X and RSWL are highly correlated, which will cause a collinearity problem in a regression model. To address this problem, two separate models (one only including the technology variable ASDE-X and another one only including the technology variable RWSL) were developed for this study. Independent variables that were found to have low statistical significance (less than 75%) were removed in a sequential variable-elimination process. The results of the developed random-effects panel data model are presented in Tables 5 and 6; it can be seen that there are only four independent variables that are significantly associated with a reduction in the number of runway incursions in the U.S. These are AirportOperation_100000, RWY_RWY, RWSL Tech, and ASDE-X Tech, and their impacts are discussed in the following sections. Note that the significance of two model parameters, i.e., intercept and sigma, indicates that the model should have an intercept and that the variance of the individual effect $\alpha_i$ is significant.

**Table 5.** Results of the random effects Poisson model for panel data for RWSL technology.

|  | Estimate Std | Error | t Value | Pr (>t) |
|---|---|---|---|---|
| (Intercept) | −2.42091 | 0.44788 | −5.405 | $6.47 \times 10^{-8}$ |
| AirportOperation_100000 | 0.09830 | 0.08018 | 1.226 | 0.2202 |
| RWY_RWY | 0.12056 | 0.09002 | 1.339 | 0.1805 |
| RWSLTech | −0.76002 | 0.32653 | −2.328 | 0.0199 |
| sigma | 3.02392 | 1.90586 | 1.587 | 0.1126 |
| Sample size | | 570 | | |

**Table 6.** Results of the random effects Poisson model for panel data for ASDE-X technology.

|  | Estimate Std | Error | t Value | Pr (>t) |
|---|---|---|---|---|
| (Intercept) | −2.43117 | 0.47353 | −5.134 | $2.83 \times 10^{-7}$ |
| AirportOperation_100000 | 0.11426 | 0.11426 | 1.385 | 0.1661 |
| RWY_RWY | 0.11884 | 0.09515 | 1.249 | 0.2117 |
| ASDEXTech | −0.38954 | 0.23128 | −1.684 | 0.0921 |
| sigma | 2.45914 | 1.38745 | 1.772 | 0.0763 |
| Sample size | | 570 | | |

- Airport Annual Operation (AirportOperation_100000):

Tables 5 and 6 indicate that an increase in the airport operations rate tended to increase the number of runway incursions. Mrazova [20] also found that with an increase in traffic levels at an airport, runway incursions will continue to increase and be a major safety concern. These results are reasonable because every year, the FAA reports an increase in airport operations for airports in the United States, as well as an increase in the rate of runway incursions. Furthermore, due to legacy runway and taxiway configurations, the risk of runway incursions has increased in conjunction with the increasing traffic volume, particularly for airports designed and built before the jet age.

- Runway-to-Runway intersecting point (RWY_RWY)"

Tables 5 and 6 indicate that the total number of runway-to-runway intersecting points was found to be a significant predictor for the increase in runway incursions at the airports. This finding is consistent with the finding of previous studies. Johnson et al. [12] also confirmed that airfield geometry plays a role in incursions, reporting that airports with runway intersections have higher incidences of incursions than airports without runway intersections. Wilke et al. [21] also found that the geometric characteristics of an airport

influence the severity of runway incursions. The author confirmed that in the U.S. airfield, the rate of safety occurrences was associated with the number of conflict points, the number of runway-to-runway conflict points, and subcontractors working on the airfield.

- RWSL Technology:

Table 5 indicates that the level of severity of runway incursions decreased significantly with the installation of Runway Status Lights (RWSLs). This finding is reasonable because when runways are unsafe, RWSL technology alerts pilots and vehicle operators. The lights automatically turn red when other traffic makes it unsafe to enter, cross, or begin takeoff. The lights provide immediate, direct alerts and do not require air traffic controller insight. Pilots can also use their best judgment and abort takeoff with the help of alerts from RWSL technology when authorized to use an unsafe runway by an air traffic controller. This finding is also consistent with the findings of Schonefeld and Moller [13], who found that when mitigation technologies are properly installed and used, they have the potential to reduce runway incursions by up to 80%. The author's findings concluded that the RWSL technology at Dallas-Ft. Worth International Airport reduced incursions by up to 70%. In addition, a report to the Department of Transportation (DOT) by Bisch et al. [22] stated that surface surveillance could reduce runway incursions regardless of other contributing factors.

- ASDE-X Technology:

The results in Table 6 indicated that the use of ASDE-X technology could help reduce runway incursions with severity levels A and B. However, the significance level of ASDE-X technology is relatively low compared to that of RWSL technology. This finding is different from the findings in the literature review. Claros et al. [15] reported that despite the use of ASDE-X technology, seven of the nine airports investigated did not see a reduction in severe runway incursions. Note that this study has relatively small sample sizes of airports and years, and it only compared the incident rates in eight large hub airports and one small hub airport between 2008 and 2014.

Comparing the results in Tables 5 and 6, it can be seen that RWSL technology a has more significant impact on reducing runway incursions than ASDE-X technology. This might be because RWSLs directly alert pilots, while ASDE-X alerts air traffic controllers of potential runway conflicts, after which the controller notifies pilots. Runway incursions might still occur if the air traffic controller does not pay proper attention to the ASDE-X display screen, intervene, and relay corrective measures to pilots on time.

According to the modeling results presented in Tables 5 and 6, Figures 5 and 6 were developed to show the estimated average number of annual runway incursions at an airport given the conditions of the amount of total annual takeoffs and landings of aircraft, the total number of runway-to-runway intersecting points, and the use of different surface technologies (ASDE-X or RSWL). From Figures 5 and 6, it also can be seen that RSWL technology can help reduce more runway incursions than ASDE-X technology.

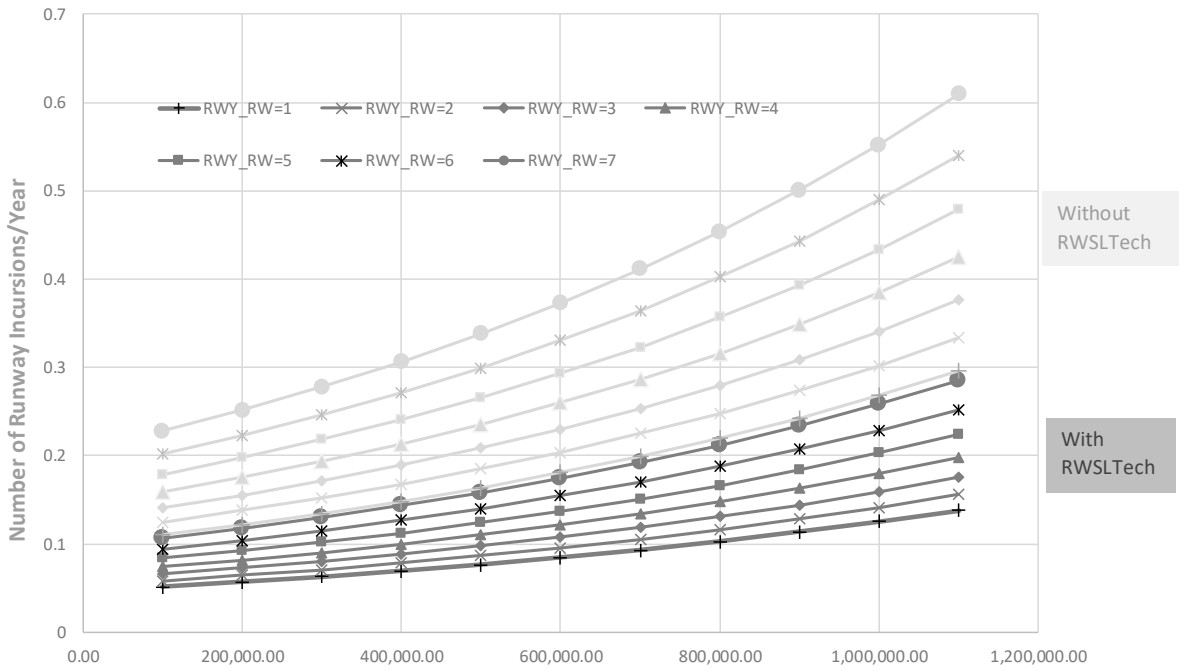

**Figure 5.** Frequency of annual runway incursions with and without RWSL technology.

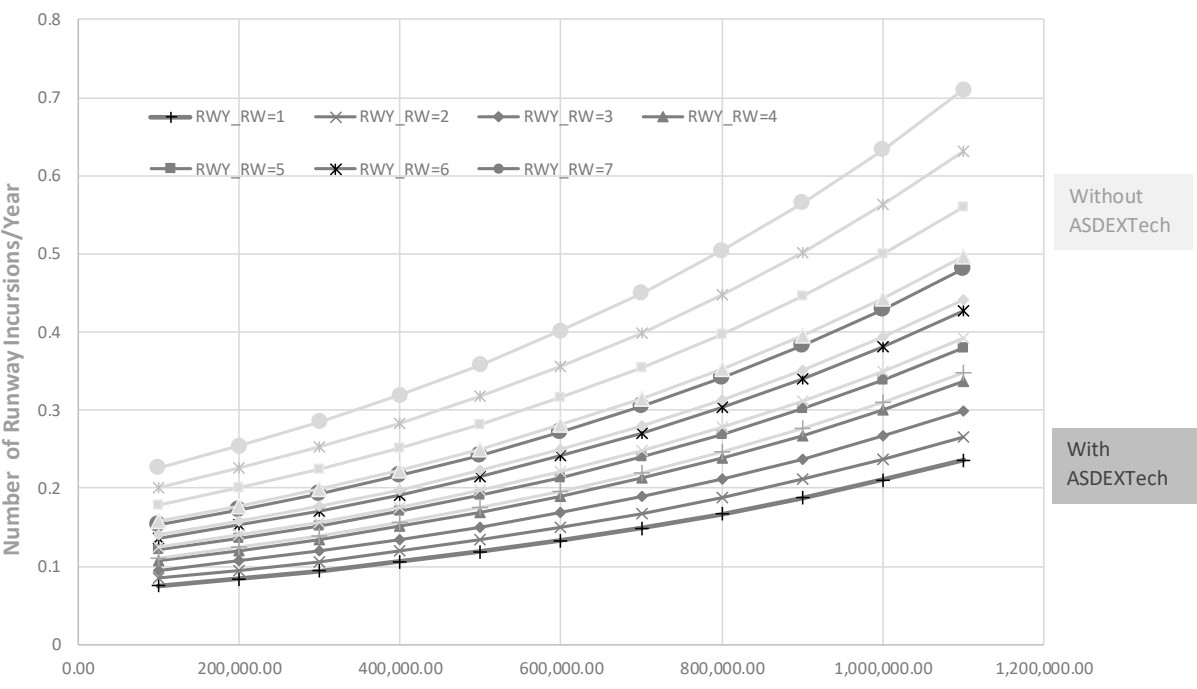

**Figure 6.** Frequency of annual runway incursions with and without ASDE-X technology.

## 6. Conclusions and Recommendations

In this research, we investigated runway incursions in 30 large hub airports in the United States based on incident data collected over 19 years from 2002 to 2020. A comprehensive set of data regarding airport layouts, mitigating technologies implemented, runway incursions, and airport operations were collected, and random effects Poisson models for panel data were developed to analyze the frequency of runway incursion incidents. The

modeling results identified the variables that have significant impacts on the occurrence of runway incursion incidents with severity levels A or B. The following items are the key findings of the study, along with some corresponding recommendations:

- ASDE-X and RWSL technologies have been proven to help reduce the rate of runway incursions. The Federal Aviation Administration (FAA) should invest and roll out more mitigating technologies to help reduce runway incursions at airports.
- RWSL technology performs better than ASDE-X technology in terms of reducing runway incursions. RWSL technology should be implemented in airports with high risks of runway incursions.
- The cramped and complex configurations of runways at airports have been shown to increase runway incursion incidents. A complicated airport design with a significant number of runway-to-runway intersection points should be avoided in future airport designs.
- For airports that currently have complex runway configurations with a significant number of runway-to-runway intersection points, the use of mitigating technologies such as runway status light (RWSLs) at conflict points is recommended to assist pilots in unsafe operations, thereby reducing runway incursions.

### 6.1. Policy Implications

The findings of this study can help FAA and aviation agencies better understand the relationship between airport geometry, runway incursions, and mitigating technologies. In addition, local governments can also allocate funding and technologies to airports with higher runway incursion rates. Overall, the results of this study can help the FAA and aviation agencies make the right decisions to fully consider technology as one of the solutions for mitigating runway incursions.

### 6.2. Limitations and Future Study Needs

There are several limitations to this study. First, this study only included large hub airports based on enplanements on NPIAS. In future research, a larger sample that includes medium and small hub airports in the U.S. could be considered. Second, some other unobserved factors like runway human factors, weather, and the time of the day may also contribute to the incursion rate and must be investigated in the future. Finally, this study developed a model for all types of runway incursions. In the future, individual models need to be developed to investigate the impacts of mitigating technologies on different types of incursions (i.e., operational incidents, pilot deviations, and vehicle/pedestrian deviations).

**Author Contributions:** Conceptualization, O.O., Q.Z. and Y.Q.; Data curation, O.O.; Formal analysis, O.O. and Y.Q.; Funding acquisition, Y.Q.; Investigation, O.O.; Methodology, Y.Q. and O.O.; Project administration, Q.Z.; Resources, Y.Q.; Supervision, Y.Q., D.O., M.A. and Y.W.; Validation, O.O. and Y.W.; Visualization, O.O.; Writing—original draft, O.O., Q.Z. and Y.Q.; Writing—review and editing, Y.Q., D.O., M.A. and Y.W. All authors have read and agreed to the published version of the manuscript.

**Funding:** This research was funded by partially by the U.S. Department of Transportation (USDOT), grant number 69A3551747133.

**Institutional Review Board Statement:** Not applicable.

**Informed Consent Statement:** Not applicable.

**Data Availability Statement:** The data presented in this study are available on request from the corresponding author. The data are not publicly available due to institutional restrictions.

**Conflicts of Interest:** The authors declare no conflict of interest.

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
