# Peer review of "Investigating Runway Incursion Incidents at United States Airports"

_futuretransp, doi:10.3390/futuretransp3040066_

Round 1

Reviewer 1 Report

The manuscript presented a macroscopic analysis of factors leading to increased runway incursions. The topic is certainly relevant and of interest, but the conclusions do not seem to present any new findings, just a confirmation of existing assumptions. If there is supposed to be a follow-up study, I would recommend that the authors provide an expanded section on 'Future Work'.

My other comments are:

1. It does not really make sense to compare 'AirportOperation' (total annual takeoffs and landings) with number of incursions, as a positive relation is almost guaranteed. Perhaps a more insightful comparison is between AirportOperation and incursion rate (incursions normalised by AirportOperation), or between AirportOperation normalised by airport capacity and number of incursions.

2. It will be helpful to include in the results some plots for visualizing the results of the model

3. Please also explain the significance of the rows 'intercept' and 'sigma' in Tables 5 and 6, and also the significance of the t-value and Pr(>t) columns.

Reviewer 2 Report

 The subject discussed in the article is important in the era of increasing air traffic density and significant for transportation safety. Generally, the paper is organized correctly. It contains the main parts of an original science paper. Nonetheless, I have got several remarks regarding the paper. Below the most important ones are specified.

The propose of the method presented by authors seems to be suitable and efficiency for studying of runway incursion incidents.  However, the authors should be present more information about  used the software.

Figure 1. The lack of the description of axes and units. The data placed on the ordinate axis of the graph should be corrected.

Tables 1 and 6 should be not separated in two pages.

There is no Poisson regression plot showing the figures used in the paper.

Line 187: The formula is Poisson regression so it is a fundamental law, but if the author wants to quote [Green 2000] he should put it in the references section.

The data in tables 5 and 6 ambiguously refer to the author's conclusions presented in lines 314 - 392. It is advisable to show the results of the calculations on the graphs.

In my opinion, conclusions are too general and obvious.

Reviewer 3 Report

In this paper the authors investigate how runway incursion mitigation technologies and runway geometric characteristics contribute to the number of runway incursions at an airport and particularly at US airports.

My comments are following:

1.       Introduction and background sections need more elaboration. Journal papers by the research community and studies- research reports made by AIRBUS, EUROCONTROL, ACI, FSF, CANSO and EASA must at minimum be included for a thorough coverage of the subject.

2.       There is not enough rational for why these two research questions were selected especially when it comes to human factors issues. There is a lot of research literature on human factors in the ATC that can be included.  A thorough examination and elimination process as to why the authors end up in these questions and excluded others is needed.

3.       The Methodology section is good.

4.       The results section needs more details.

5.       There is a clear need for a proper and a well-developed discussion and a conclusion section. This is the weak point of the paper. It doesn’t provide convincing discussion and conclusion sections.

6.       Page 3, Section 2 (Background), line 3, “For example, in 1997” must be substituted with “For example, in 1977” because the Tenerife accident happened in 1977.

Overall, the paper has the potential to  improve subject to a major revision.

Reviewer 4 Report

Dear authors, I had the pleasure to review your manuscript. I include more details on my observations in the attachment.

You picked an interesting and relevant topic, i.e. runway safety risk / incursion risk. My main criticism is the structure of the paper (or storyline) that can be improved. In particular, rethink how to better present and separate the background, method/data, and modelling work. This will improve your paper significantly.
I think all the pieces are there, just make sure to not intermingle things to heavily.

Congrats to the work. And I hope that my observations will help to improve your paper!

The comments/observations also contain recommendations in terms of language. 

I strongly recommend that you scope the use of the (safety) terminology and apply it consistently across the paper. This will help to strengthen the relevance of the paper for aviation (safety) experts and decision-makers, and other researchers.
Have a careful read and try to eliminate subsequent sentences "more or less" saying the same. 

Round 2

Reviewer 3 Report

The authors addressed adequately all my comments and I am satisfied with the revision.